# Synthesis and Properties of Photocurable Polymers Derived from the Polyesters of Glycerol and Aliphatic Dicarboxylic Acids

**DOI:** 10.3390/polym16091278

**Published:** 2024-05-02

**Authors:** Rui Hu, Weipeng Yao, Yingjuan Fu, Fuyuan Lu, Xiaoqian Chen

**Affiliations:** State Key Laboratory of Biobased Material and Green Papermaking, Qilu University of Technology (Shandong Academy of Sciences), Jinan 250353, China; hrui199909@163.com (R.H.); ywp84302@163.com (W.Y.); lufy5321@163.com (F.L.); chenxiaoqian@qlu.edu.cn (X.C.)

**Keywords:** photocuring, acrylated poly(glycerol-*co*-tetradecanedioic acid), acetylation modification, acryloyl chloride, elastomers

## Abstract

The rapid development of 3D printing technology and the emerging applications of shape memory elastomer have greatly stimulated the research of photocurable polymers. In this work, glycerol (Gly) was polycondensed with sebacic, dodecanedioic, or tetradecanedioic acids to provide precursor polyesters with hydroxyl or carboxyl terminal groups, which were further chemically functionalized by acryloyl chloride to introduce sufficient, photocurable, and unsaturated double bonds. The chemical structures of the acrylated polyesters were characterized by FT IR and NMR spectroscopies. The photoinitiated crosslinking behavior of the acrylated polyesters under ultraviolet irradiation without the addition of any photoinitiator was investigated. The results showed that the precursor polyesters that had a greater number of terminated hydroxyls and a less branched structure obtained a relatively high acetylation degree. A longer chain of aliphatic dicarboxylic acids (ADCAs) and higher ADCA proportion lead to a relatively lower photopolymerization rate of acrylated polyesters. However, the photocured elastomers with a higher ADCA proportion or longer-chain ADCAs resulted in better mechanical properties and a lower degradation rate. The glass transition temperature (*T_g_*) of the elastomer increased with the alkyl chain length of the ADCAs, and a higher Gly proportion resulted in a lower *T_g_* of the elastomer due to its higher crosslinking density. Thermal gravimetric analysis (TGA) showed that the chain length of the ADCAs and the molar ratio of Gly to ADCAs had less of an effect on the thermal stability of the elastomer. As the physicochemical properties can be adjusted by choosing the alkyl chain length of the ADCAs, as well as changing the ratio of Gly:ADCA, the photocurable polyesters are expected to be applied in multiple fields.

## 1. Introduction

Photocuring refers to a curing process of a monomer, oligomer, or polymeric matrix under light induction, where it transforms liquid matter into solid polymer network [1]. Compared to traditional thermal curing, photopolymerization technology has the advantages of rapid curing speed, high efficiency, low energy consumption, no/less solvent emission, and environmental friendliness [2,3], and it is in great demand in many fields such as coatings [4,5], inks [6], adhesives [7], biomedical engineering [8,9,10,11], electron devices [12], as well as in the construction industry, automobile industry, and aerospace industry [13]. The success of photocuring critically depends on the choice of the photopolymer, photoinitiator, and photoabsorber used [10]. According to reaction mechanisms, the photocuring system can be divided into photoinitiated free radical polymerizations and light/radiation-induced cationic polymerizations [2,14]. The former mainly includes (methyl)acrylates [15,16] and the mercaptan/polyene system [17], while the latter mainly refers to the epoxide and vinyl ether type. During the photocuring process, the photoinitiators produce free radicals or active cations under light radiation. Then, these free radicals or active cations induce a polymerization of the prepolymer (oligomer) or monomer, which form covalently cross-linked polymer networks by chain reaction [18]. The photoinitiated free radical polymerization has played a seminal role in a variety of practical industrial applications, and the acrylate system is one of the most common free radical photocuring systems, which is usually composed of oligomers or prepolymers containing acrylate functional groups, acrylate monomers, and free radical photoinitiators [3,19]. For example, two photocrosslinkable polymers, poly(ethylene glycol) dimethacrylate, and poly(ethylene glycol) diacrylate have been used as primary materials to prepare the photoreactive solutions with which the multi-material scaffolds were successfully fabricated [20]. 

While the field of free radical photocuring is well established, the types of functional monomers/oligomers available for photocuring are not rich enough compared with the traditional curing technology [1]; thus, it is difficult to prepare high-performance materials with specific functions for various application areas. And the defective properties of photopolymers restrict their development and application in both industry and daily life [21]. As the commonly used photosensitive substances are mostly bifunctional or multi-functional monomers/oligomers [22], the vinyl end-capped modification of functional monomers/oligomers is an efficient strategy for obtaining target photocurable materials [23]. The commonly used photoreactive end groups generally contain unsaturated double bonds, such as (meth)acrylate [24,25], acryloyl [26], and acrylamide [27], which enable light-induced chemical crosslinking. As the (meth)acrylate-containing monomers/oligomers possess an acrylate double bond of high reactivity in radical polymerization [3], introducing the (meth)acrylate groups to the functional monomers/oligomers is preferred. By means of the methacrylation of chitosan with methacrylic anhydride (MA), a photocurable chitosan-based bioink was developed for digital light processing (DLP) three-dimensional (3D) printing, which provided a promising approach to obtain customized bolus for radiotherapy application [9]. Si et al. [28] prepared a methacrylate-functionalized polydimethylsiloxane (MA-PDMS) via modifying the polydimethylsiloxane with photoresponsive groups (methacrylate groups). The UV-induced curing of the methacrylate-terminated PDMS could complete within 30 s, thereby indicating that its crosslinking rate was three orders of magnitude larger than that of conventional thermal crosslinking. Şabani et al. [29] synthesized hyperbranched urethane acrylates (UA/HB-Pes) by modifying the hydroxyl groups of hyperbranched polyester polyols with urethane acrylates. It was found that the formulations containing the unsaturated oligomers UA/HB-PEs, reactive diluents, and photoinitiators could successfully polymerize under UV irradiation, and the resultant UV-cured films had good adhesion and high gloss properties. In view of the high reactivity of acryloyl chloride, Liu et al. [30] used acryloyl chloride to modify the calcium sulfate whiskers and particles that were pre-coated with chitosan, thereby endowing the particle surfaces with carbon–carbon double bonds. The modified inorganic fillers enhanced the tensile strength and impact strength of the cured photosensitive resin. In addition, modification of the functional monomers/oligomers is an effective way through which to improve the properties of photocrosslinkable materials. For instance, photocuring inks based on acryloyl-modified polyethylene glycol exhibit not only high tensile strength and elongation, but also excellent resilience [21]. 

Recently, with the enhancement of environmental awareness and the increasingly strict control of organic volatile components, developing various high-performance and multifunctional photosensitive polymers that meet the requirement of 5E (efficient, enabling, economical, energy-saving, and environmentally friendly) for photocuring technology has become a key work in all of the application fields [31]. Polyglycerol dicarboxylic acid ester is a promising class of thermosetting elastomer that is synthesized via the straightforward polycondensation of glycerol and aliphatic dicarboxylic acids such as sebacic acid and dodecanedioic acid [32,33,34,35]. Due to its inherent advantages, such as biocompatibility, biodegradability, non-toxic degradation products, elastomeric mechanical properties, and shape-memory behavior, the polyglycerol dicarboxylic acid ester has been identified as an attractive candidate material for tissue engineering scaffolds, medical devices, and drug delivery [32,34,36,37]. Furthermore, the cross-linked network of the polyglycerol dicarboxylic acid ester can be used as a phase transition material due to its temperature-dependent light scattering properties. For instance, poly(glycerol-dodecanoate) has been used as a transparency-tunable and impact-resistant coating material for windows due to its behavior of being transformed from a translucent state to a transparent state when the temperature rises above its glass transition temperature (*T*_trans_) [38]. However, the harsh preparation conditions (such as high curing temperature and long reaction time) of the polyglycerol dicarboxylic acid ester limit its potential application range [39]. One convenient strategy to overcome the limitations of the thermal curing process is endowing the polyglycerol dicarboxylic acid ester with photoreactivity and photocrosslinking. This works because the photocurable double bonds can be easily incorporated into its backbone. Yeh et al. [40] synthesized two photocurable acrylated poly(glycerol sebacate) macromers with varied molecular weights and viscosity, which were confirmed as promising materials for the fabrication of elastomeric biomedical scaffolds. Akman et al. [41] developed a UV curable polymer via the acrylation of poly(glycerol dodecanedioate), which could be used to 3D print biodegradable shape memory samples and shows promise as a novel material for biomedical applications. Moreover, the thermal, mechanical, and chemical properties of the photocured materials can be easily controlled by varying the degree of acrylation [39,41]. 

Currently, the rapidly growing field of 3D printing technology and the emerging applications of shape memory polymer networks have greatly stimulated the research of photocurable polymers [18]. The incorporation of photocurable chemistries into functional materials is a viable method to ensure the rapid crosslinking of polymer networks while retaining their intrinsic properties [41]. Although some studies have been conducted on the vinyl modification of the polyglycerol dicarboxylic acid ester [39,40,41], the relevant research is still in its infancy. In particular, how the prepolymer structures such as the carbon chain lengths of dicarboxylic acids and the ratio of glycerol to dicarboxylic acid affect the acrylation efficiency of the prepolymer and the photocuring behavior of the acrylated polymers is still unclear. The present work describes the use of acryloyl chloride to react with the hydroxyl groups in polyesters that are synthesized via polycondensation of glycerol (Gly) and aliphatic dicarboxylic acids (ADCAs) for introducing photoreactive double bonds into the backbone in order to satisfy the rapid photocuring process. The synthesized polymers were characterized by FT IR, NMR, and GPC. The photocuring performance of the acrylated polyesters was evaluated, and the effects of the alkyl chain lengths and the mole ratio of Gly to ADCAs on the curing rate, mechanical properties, and degradation properties of the photocured networks were investigated. 

## 2. Materials and Methods

### 2.1. Materials

Glycerol (Gly), sebacic acid (SA), dodecanedioic acid (DA), and tetradecanedioic acid (TA) were purchased from Sigma-Aldrich (Shanghai, China). Acryloyl chloride (AC) was purchased from Shanghai Aladdin Bio-chemical Technology Co., Ltd. (Shanghai, China). Triethylamine (TEA) and ethyl acetate (EA) were purchased from Shanghai Macklin Biochemical Co., Ltd. (Shanghai, China) All chemicals were used as received without further purification.

### 2.2. Synthesis of Polyglycerol Dicarboxylic Acid Esters

The polyesters of glycerol (Gly) and aliphatic dicarboxylic acids (ADCAs), i.e., poly(glycerol-*co*-sebacic acid) (PGS), poly(glycerol-*co*-dodecanedioic acid) (PGD), and poly(glycerol-*co*-tetradecanedioic acid) (PGT) were synthesized according to the reported method with some modifications [34,42,43]. Briefly, anhydrous Gly and ADCAs with the molar ratio of 1:1 were added into a 250 mL three-neck flask. The mixture was heated up to their melting temperature by an oil bath, and then was kept at 120 °C for 24 h under a continuous magnetic stirring and nitrogen environment. In order to remove the by-product water, nitrogen bubbles were continuously inserted into the reaction mixture to carry away the water vapor. The resulting prepolymers (PGS, PGD, and PGT) were transferred to glass vials for further analysis and use. 

### 2.3. Acetylation of the Polyglycerol Dicarboxylic Acid Esters

The prepolymers (polyglycerol dicarboxylic acid esters) were acrylated with acryloyl chloride by dissolving the prepolymers in dichloromethane using trimethylamine as the acid acceptors. Briefly, the PGD prepolymer (5 g, with 0.0388 mmol of hydroxyl groups) was dissolved in 20 mL of anhydrous dichloromethane containing an equimolar amount of triethylamine. After the prepolymer was completely dissolved, the reaction solution was refrigerated to 0 °C for 10 min. Then, acryloyl chloride (approximately 1.0 mol/mol of hydroxyl groups on the PGD prepolymer) was added dropwise using a constant pressure dropping funnel to the reaction solution. After that, the mixture was warmed up to reach room temperature and stirred till it reacted for an additional 12 h. The reaction mixture was evaporated by a rotary evaporator to remove the dichloromethane. The remaining liquid was dissolved in ethyl acetate, and it was then centrifuged to remove the salt. After the ethyl acetate was removed using a rotary evaporator, the obtained viscous liquid (acetylated polyesters) was refrigerated and preserved [39]. The acetylated polyesters were named PGSA, PGDA, and PGTA according to the precursor polymers, i.e., poly(glycerol-*co*-sebacic acid) (PGS), poly(glycerol-*co*-dodecanedioic acid) (PGD), and poly(glycerol-*co*-tetradecanedioic acid) (PGT), respectively.

### 2.4. Photocuring of the UV-Curable Polymers 

A certain amount of acetylated polyester was poured into a rectangular silicone mold with a size of 10 mm × 30 mm × 5 mm, where the polymerization reaction was initiated by ultraviolet light (365 nm, ca. 10 mW/cm^2^, model ZF-1). After the photopolymerization was carried out for a scheduled time at room temperature, the photocured solid network (elastomer) was immersed in dichloromethane to remove the uncured molecules. Then, the undissolved solid was taken out and dried in an oven for 6 h at 45 °C. The photocuring degree was calculated according to the mass differential method. The photocured elastomer were named pc-PGSA, pc-PGDA, and pc-PGTA according to the acetylated polyesters PGSA, PGDA, and PGTA, respectively.

### 2.5. Characterization of the Polymers and Photocured Elastomers

#### 2.5.1. Fourier-Transform Infrared Spectroscopy (FT IR) Analysis

The FT IR spectra of the polymers and photocured elastomers were obtained from a spectrophotometer (Prestige-21, Shimadzu, Japan) using the KBr-pellet method. Each spectrum was recorded over 10 scans in the frequency range of 4000 to 500 cm^−1^ with a resolution of 2 cm^−1^. The ratio of primary and secondary hydroxyl groups (*p*-OH/*s*-OH) was obtained by calculating the absorbance ratio between 1048 cm^−1^ and 1099 cm^−1^ [43].

#### 2.5.2. NMR Analysis

The NMR analysis of the polymers (where a 60 mg of sample was dissolved in 0.5 mL of DMSO-d6) was performed at room temperature by an AVANCE II 400 spectrometer (Bruker, Karlsruhe, Germany) equipped with a 5 mm broadband probe and a gradient field in the Z-direction. The ^1^H NMR spectrum was recorded on the spectrometer under the condition of a minimum of 8 scans, a sweep width of 400 MHz, an acquisition time of 2.0 s, and a relaxation delay time of 3 s, while the ^13^C NMR spectrum was acquired with a minimum of 20,000 scans, a sweep width of 400 MHz, an acquisition time of 0.4 s, and a relaxation delay time of 1.5 s. 

#### 2.5.3. GPC Analysis

The average molecular weight (*Mw*), number of the average molecular weight (*Mn*), and polymer distribution index (*PDI* = *Mw/Mn*) of the polymers were determined by gel permeation chromatography (GPC) on an Agilent PL-GPC50 instrument using tetrahydrofuran as the eluent. The concentration of the sample was 0.1 mg mL^−1^, and the injection volume was 100.0 μL. The polystyrene standards were used for calibration.

#### 2.5.4. Differential Scanning Calorimetry (DSC) and Thermal Gravimetric (TGA) Analysis

The thermal property of the photocured elastomers was characterized using a differential scanning calorimeter (DSC Q20, TA Instruments) under a nitrogen atmosphere. The sample (10 mg) was initially heated from room temperature up to 90 °C at a rate of 10 °C min^−1^. After being held at 90 °C for 3 min, the sample was cooled from 90 °C down to −50 °C, held at −50 °C for 3 min, and then re-heated up to 70 °C using a cooling/heating rate of 10 °C min^−1^. During the re-heating scan, the glass transition temperature (*T_g_*) was determined from the maximum of the endothermic peak [44]. 

Thermal gravimetric analyses (TGAs) of the elastomers were carried out on a simultaneous thermal analyzer (STA 449 F3, Netzsch, Germany) to determine the decomposition temperature. Approximately 10 mg of the pre-vacuum-dried samples were heated from ambient temperature up to 600 °C at a heating rate of 10 °C/min under a nitrogen atmosphere. The high-purity nitrogen (99.999%) was used as a carrier gas with a flow rate of 10 mL/min.

#### 2.5.5. Mechanical Property Analysis

Tensile tests were performed using a texture analyzer (TA.TX PlusC, Stable Micro Systems, UK) with a 300 N load cell for estimating the tensile properties of the cured elastomers. The precise width and thickness of the specimen were measured prior to testing, and the testing speed was 12.0 mm min^−1^. All of the tensile tests were conducted at 37 °C and were continued until the specimens fractured. The Young’s modulus was calculated from the initial slope of the tensile stress–strain curve, and the toughness was calculated from the integral area under the curve. The strain at break was defined as the highest strain value prior to the fracture of the sample.

#### 2.5.6. Degradation Analysis 

The degradation rates of the photocured elastomers were evaluated by the mass differential after incubation of the samples in phosphate-buffered saline (PBS) (pH = 7.4) for 7 d, 14 d, 21 d, and 28 d. In brief, the dry samples with a certain mass were cut into small pieces of approximately the same size and immersed in 10 mL of PBS for certain scheduled times at 37 °C. After being taken out from the PBS, the residual solids were rinsed thoroughly with deionized water, dried at 45 °C for 6 h, and then weighed. The PBS was changed every 72 h to maintain the degradation pH.

## 3. Results and Discussion

### 3.1. Effect of the Mole Ratio and Chain Lengths of ADCAs on the Prepolymers

As shown in Figure 1a, the aliphatic dicarboxylic acids (ADCAs) with the alkyl chain lengths of 8, 10, and 12, i.e., sebacic acid (SA), dodecanedioic acid (DA), and tetradecanedioic acid (TA), respectively, were used to react with glycerol (Gly) to produce precursor polyesters (prepolymers). As expected, the average molecular weights and polydispersity indexes (*PDI*) of the synthesized prepolymers increased with the polycondensation times (Table 1). As Gly is a trifunctional monomer with two primary hydroxyl groups (–OH) and one secondary –OH, the polycondensation of Gly and ADCAs results in prepolymers composed of repetitive units, whose structures are similar to mono-, di-, and triglycerides [45]. Compared to the primary –OH in Gly, the secondary –OH with higher steric inhibition was less reactive [44]; therefore, in the early reaction stage, the synthesized prepolymers had a predominantly linear structure due to the superior reactivity of primary –OH [43]. With an extension of the reaction time, the secondary –OH might react with the ADCAs, thereby producing a branched prepolymer with a higher *PDI*. This could be confirmed by the increased esterification degree of secondary –OH (ED_2_), as shown in Table 1. Moreover, the longer alkyl chain lengths of the ADCA units led to the higher molecular weights of the prepolymers. Table 1 also shows that the molecular weights and polydispersity indexes of the prepolymers increased with the increase in the proportions of the ADCAs. The greater the number of ADCAs that exist in the reaction system (Gly:ADCA = 1:1.5), the greater the chance they could have to react with the secondary –OH, which would lead to the production of more branched molecules and thus a higher *PDI* of the prepolymer. The significant increase in ED_2_ and carbonyl group content of the prepolymers alongside a decrease in the molar ratio of Gly:ADCA confirmed this hypothesis (Table 1).

Since the residual hydroxyl groups in the prepolymers are the preferred sites of acrylation, the acrylated secondary –OHs act as the main crosslinking sites, and the *p*-OH/*s*-OH of the prepolymers were thus evaluated. As shown in Table 1, including longer-chain ADCAs, as well as prolonging the polycondensation time, resulted in a decrease in the *p*-OH/*s*-OH of the prepolymers. This is because the ADCAs with a longer alkyl chain tend to react with the primary –OHs rather than the secondary –OHs of the Gly due to their greater steric hindrance. In addition, more ADCAs in the reaction system consumed more of the hydroxyl groups of the Gly, thus leading to a low *p*-OH/*s*-OH when the molar ratio of Gly:ADCA reduced to 1:1.5. 

### 3.2. Acetylation Efficiency of the Prepolymers

As the acrylic double bond can be activated to generate a polymer or a cross-linked network by ultraviolet irradiation or heating [46], the photocurable polymers with pendant acrylate groups were synthesized by using acryloyl chloride as the acrylating reagents and triethylamine as the deacid reagents in mild reaction conditions (Figure 1a). In a view to obtain as much of an acylation degree (especially the acylation of the secondary –OH) as possible, the prepolymers that were prepared under 120 °C for 24 h were chosen. The FT IR spectra of the acetylated polyesters were compared with that of the prepolymers, as shown in Figure 1b. In the spectra of the prepolymers, the characteristic absorption peaks included hydrogen-bonded hydroxyl groups (3442 cm^−1^), alkyl C-H stretching vibration (2922 and 2853 cm^−1^), C=O stretching vibrations of ester bonds (1738 cm^−1^), carboxylic acid O-H bends (1388 cm^−1^), the C-O of ester bonds (1170 cm^−1^), secondary saturated alcohol groups (1099 cm^−1^), and primary saturated alcohol groups (1048 cm^−1^) [43]. For the acetylated polyesters, the appearance of the peaks at 1635 cm^−1^ and 810 cm^−1^ were attributed to the stretching vibration of the acryloyl C=C and the out-of-plane bending vibration of acryloyl =C-H, which demonstrated the successful acetylation of the perpolymers by acryloyl chloride [46]. Moreover, the near disappearance of the characteristic peaks of hydroxyl groups at 3442 cm^−1^ in the spectra of the acetylated polyesters indicated that most of the active hydroxyl groups in the prepolymer molecules were replaced by the acrylate groups.

The incorporation of acrylate groups to the polyesters was further confirmed by NMR analysis. The ^1^H NMR spectra of the polymers (Figure 2a) displayed typical proton signals in the polyesters of Gly and ADCAs, such as the protons from the aliphatic chains of the ADCA units (δ_H_ 1.21 ppm (c), δ_H_ 1.48 ppm (b), and δ_H_ 2.21 ppm (a)), as well as the protons from the Gly units (δ_H_ 3.0–3.4 ppm (d), δ_H_ 3.8–4.3 ppm (e), and δ_H_ 4.9–5.3 ppm (f)) [44]. Compared to the prepolymers, the proton signals of the –CH=CH_2_ in the acetylated polyesters appeared at δ_H_ 5.94, 6.16, and 6.31 ppm (annotated as *i*, *h*, and *g*, respectively) in the spectra, thereby indicating that the acrylate groups had been introduced into the polymers [39]. Moreover, the greatly reduced intensity of proton signals at δ_H_ 3.0–3.4 ppm (d) that corresponded to the –CH and –CH_2_ connected with the unreacted –OH was clearly discernible, thus further confirming the acrylation of residual hydroxyl groups in the Gly units by acryloyl chloride.

In the ^13^C NMR spectra of the polymers (Figure 2b), the signals observed at δ_C_ 70.1 ppm (C_1_), and δ_C_ 64.9 ppm (C_2_) were ascribed to the alkyl skeleton of the Gly units, while the three sharp signals at δ_C_ 33.9 ppm (C_6_), δ_C_ 29.3 ppm (C_8_), and δ_C_ 24.9 ppm (C_7_), respectively, were attributed to the alkyl carbon chains of the ADCA units. The signals at approximately δ_C_ 174 ppm (C_9_) were assigned to the carbonyl groups that originated from esterification of the hydroxyl groups in the Gly with the carboxyl groups in the ADCAs. The acetylated polyesters evidently exhibited new signals at δ_H_ 130.8 ppm (C_4_), and δ_H_ 128.0 ppm (C_5_) belonged to C=C. In addition, the ones at δ_H_ 165.7 ppm (C_3_) were ascribed to the carbonyl groups related to the acrylate groups.

The integration of the proton peak in the ^1^H NMR spectroscopy was used to calculate the acrylation percentage (Acr%) of the polymer [41]. From the ^13^C NMR spectroscopy, the ratio of carbonyl groups was obtained by comparing the relative intensity of the signals at 164.5–167.5 ppm with the ones at 62.5–72.5 ppm. According to the acetylation chemistry, the acrylation degree increased with the molar ratio of the acryloyl chloride and reaction time [39,41]. In order to achieve the most similar acetylation degree as possible, excessive acryloyl chloride was used for all of the prepolymers during the reaction process. Figure 3 shows that all of the acetylated polyesters had a relatively high acetylation degree due to the sufficient amount of residual hydroxyl groups in the prepolymers endowing a greater number of reactive sites. The acetylated polyesters with the same molar ratio of Gly:ADCA = 1:1 exhibited a relatively similar Acr% except PGDT, which had the lowest acylation degree due to the highest steric hindrance effect of the long carbon chains. Moreover, the Acr% of the acetylated polyesters slightly decreased when the molar ratio of Gly:ADCA decreased. This was consistent with the results of the ^13^C NMR spectra, where the carbonyl groups at δ_H_ 165.7 ppm (C_3_) decreased with the ADCA proportion. This could be attributed to the fact that more of the hydroxyl groups in Gly were esterified by ADCAs during the synthetization of the prepolymers (PGD_1:1.5_), which decreased the acetylation active sites. Conversely, the PGS_1:0.8_ and PGD_1:0.8_ that had more terminated hydroxyls and less of a branched structure showed higher Acr%. 

### 3.3. Photocuring Performance of the Acetylated Polyesters 

In considering the application of photocurable polymers in the fields of medicine and bioengineering, which generally forbid the presence of other adulterated substances, no photoinitiator was used when performing photocuring. The irradiation of the acetylated polyesters with an ultraviolet light of 365 nm led to a cleavage of unsaturated C=C bonds that went on to form free radicals, which initiated the copolymerization of the polymer molecules. This was confirmed by the remarkable increase in the peak at 2922 cm^−1^ (Figure 1c), which corresponded to the alkyl C-H stretching vibration [39]. The characteristic peaks of the unsaturated double bonds at 1635 cm^−1^ and 810 cm^−1^ produced a significant attenuation, thus further indicating that the polymer was successfully cured after ultraviolet light irradiation. The cross-linked long polymer chains formed solid networks, which were undissolved in solvent dichloromethane; meanwhile, the uncured molecules showed a propensity to dissolve in the solvent after a long period of soaking extraction. Thus, the gel content was used to characterize the cross-linked degree of the polymers [47]. As shown in Figure 4, all of the acetylated polyesters needed a longer time to photocure completely due to no photoinitiator being used. And it was also found that the oxygen inhibition effect reduced the photopolymerization rates of the polymers under an air atmosphere [18]. 

It was found that the acetylated polyesters exhibited similar photocuring curves, where a high photopolymerization rate appeared in the early curing stage. The surface polymer molecules will preferentially absorb the UV light to photopolymerize and form a cured layer. Once the surface cured layer is formed, the transmission of the UV light might be hindered, and the internal polymer molecules will not be able to receive enough light, which would limit the production of the internal free radicals and result in a reduction in the curing speed [48]. Moreover, the fast conversion of unsaturated double bonds might cause the gel effect to appear prematurely [46], which will make the chain segments move with difficulty and increase the steric effect between the molecules, thus limiting the photopolymerization efficiency of the unsaturated double bonds. In addition, as the photocrosslinking of the polymers increased, the initiation and propagation of the polymerization became diffusion-controlled; thus, the formed polymer network reduced the diffusion rate of the free radicals, thereby leading to a decrease in the polymerization rate [18]. Figure 4 also shows that, due to the steric hindrance, for the PGTA, the longer-chain ADCAs limited the diffusion and migration of the primary living free radicals, thereby leading to a relatively low rate of photopolymerization. Moreover, the restricted movement of the unsaturated double bond by the ADCA segment also hindered the complete curing of the PDTA [46]. Additionally, when the molar ratio of Gly:ADCA decreased, the polymerization efficiency of the polymers slightly decreased, which was attributed to the relatively high molecular weight (Table 1), as well as the lower Acr% of the acetylated polyesters (such as cp-PGDA_1:1.5_). 

### 3.4. The Properties of Photocured Elastomers

The photoinitiated radical polymerization of the acetylated polyesters yielded a kind of solid network, where the crosslinked polymer chains formed an amorphous domain, and the un-crosslinked molecules formed a semicrystalline domain [38]. The DSC analysis (Figure 5) showed that a higher molar ratio of Gly:ADCA resulted in a lower *T*_g_ in the pc-PGDA_1:0.8_, while the elastomers with a lower Gly:ADCA molar ratio exhibited a higher *T*_g_. This indicated that the lower molecular weight and higher Acr% of the PGDA_1:0.8_ that was obtained from a molar ratio of Gly:ADCA = 1:0.8 yielded a corresponding photocured elastomer with a higher crosslinking density, thereby reducing the flexibility of the molecular chain and preventing the formation of semicrystalline domains, which resulted in a relatively lower *T*_g_. When the molar ratio of Gly: ADCA was the same (1:1), the *T*_g_ of the photocured elastomers increased with the alkyl chain length of the ADCAs. For example, the *T*_g_ of the pc-PGDA with ten CH_2_ in the ADCAs was 11.5 °C, while the pc-PGTA with twelve CH_2_ in the ADCAs showed a *T_g_* of 25.6 °C. The longer chains were less mobile, which made the polymer molecules more resistant to crosslinking but more amenable to forming semi-crystalline regions [41], thus resulting in increased *T*_g_ values in the photocured elastomers.

As shown in Figure 5d,e, all of the photocured elastomers showed almost the same thermal decomposition curve, and the temperatures (*T_m_*) corresponding to the maximum decomposition rates were very similar. However, the longer molecular chain of the ADCAs resulted in a slightly higher *T_m_* (422.4 °C) in the cp-PGTA. And it was also found that a higher cross-linking density in the cp-PGSA and cp-PGDA_1:0.8_ also led to slightly higher *T_m_* (Figure 5e).

The mechanical properties of the photocured elastomers were evaluated by tensile testing. The representative stress–strain curves are shown in Figure 6, where a nonlinear elastic mechanical performance is displayed. The stress and strain at the point of break of the photocured elastomers pc-PGDA increased with increasing the ADCA proportion in the polymers. For instance, when the Gly:ADCA molar ratio decreased from 1:0.8 to 1:1.5, the stress at break (i.e., the ultimate tensile strength) increased from 0.45 MPa up to 1.73 MPa; meanwhile, the strain at break increased from 7.10% up to 18.33% with the Young’s modulus, and toughness also increased from 10.50 MPa to 23.99 MPa and 1.95 KJ/m^3^ to 25.31 KJ/m^3^, respectively. The mechanical properties of the elastomers were a comprehensive result of the gel content, molecular chain length, and the branching and crosslinking degree of the elastomer. The a greater Gly proportion in the prepolymer led to a relatively low molecular weight and high cross-linking density of the cured pc-PGDA, which made the pc-PGDA become brittle and less tough [49].

Figure 6c also shows that the stress and strain at the break of the pc-PGDA were significantly higher than that of the pc-PGSA and pc-PGTA. And it was also found that the pc-PGTA showed a relatively lower strain at break compared to the pc-PGSA. This meant that longer alkyl chain enhanced the strength and toughness of the elastomers, but too long molecular chains could lead to an increased rigidity of the elastomers, such as the pc-PGTA that had the highest Young’s modulus (Figure 6d). These differences demonstrate that the properties of photocured elastomers can be adjusted by changing the molar ratio of Gly:ADCA and choosing the carbon chain length of ADCAs, and this holds even apart from varying the acrylation percentage [50].

The degradation properties of the elastomers were evaluated by immersing them in PBS at 37 °C. Figure 7 shows that the degradation of the photocured elastomers was slightly faster than that of the thermal-cured elastomers (which were cured at 130 °C for 48 h), although their mass loss profile via hydrolysis were much similar. The degradation mechanism of the polyesters was the hydrolysis of the ester bonds; thus, the higher degradation rates of the photocured elastomers was attributed to the increased carbonyl groups introduced by acetylation (Figure 2b). As can be seen from Figure 7a, the mass losses of the corresponding elastomers decreased with increasing the alkyl chain length of the ADCAs. This was due to the higher steric hindrance effect of the longer alkyl chain, which hindered the cleavage of the ester bonds and resulted in reductions in the degradation rates. Moreover, the photocured elastomer with a lower molar ratio of Gly:ADCA exhibited a slower degradation rate (Figure 7b), which was due to its higher molecular weight (such as in cp-PGDA_1:1_ vs. cp-PGDA_1:0.8_).

## 4. Conclusions

A photocurable polymer was developed via the introduction of acrylate groups to the backbone of poly(glycerol-*co*-aliphatic dicarboxylic acid). More terminated hydroxyls and a less branched structure are beneficial to the acetylation modification of the precursor polyester. The acetylated polyester can photocure under the irradiation of ultraviolet light without the use of a photoinitiator. A lower molar ratio of Gly:ADCA and longer-alkyl-chain ADCAs led to a relatively lower photopolymerization rate. However, the photocured elastomer with an appropriate ADCA carbon chain length and a high ADCA proportion exhibited high mechanical properties and a low degradation rate. Moreover, the longer-chain ADCAs resulted in a higher *T_g_* in the photocured elastomer, and the elastomer with a lower Gly:ADCA molar ratio exhibited a higher *T_g_*. Therefore, the properties of the photocured elastomers can be adjusted by changing the molar ratio of Gly:ADCA, as well as by choosing the alkyl chain length of the ADCAs.

## Figures and Tables

**Figure 1 polymers-16-01278-f001:**
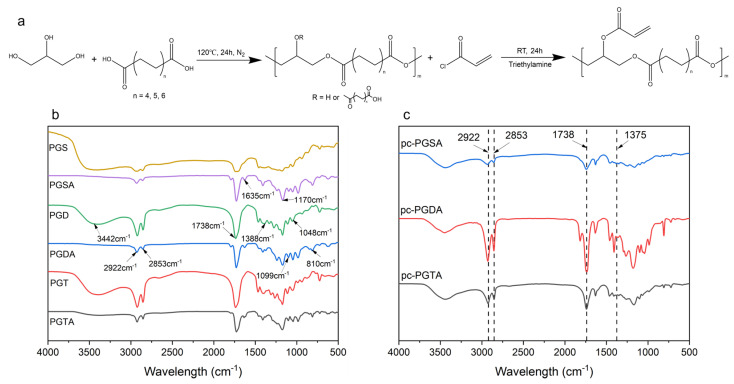
(**a**) Synthesis route of the photocurable polymers; (**b**) FT−IR spectra of the polyglycerol dicarboxylic acid esters and the corresponding acrylated polyesters; and (**c**) FT−IR spectra of the photocured elastomers photopolymerized at RT for 24 h.

**Figure 2 polymers-16-01278-f002:**
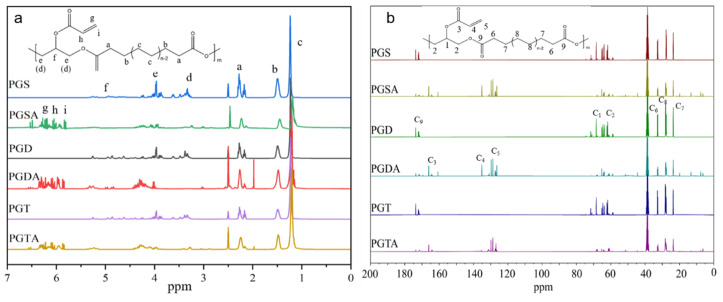
(**a**) The ^1^H NMR and (**b**) ^13^C NMR spectra of the polyglycerol dicarboxylic acid esters and the corresponding acrylated polyesters.

**Figure 3 polymers-16-01278-f003:**
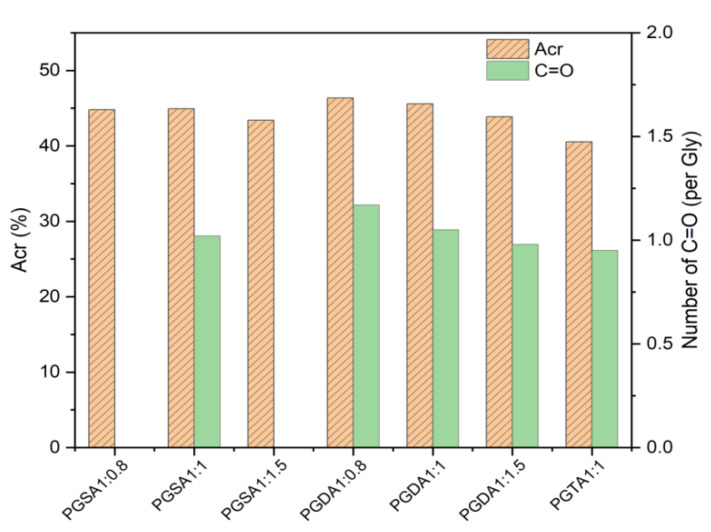
The acrylation percentage (Acr%) and number of carbonyl groups (at δ_H_ 165 ppm) of the acrylated polyesters.

**Figure 4 polymers-16-01278-f004:**
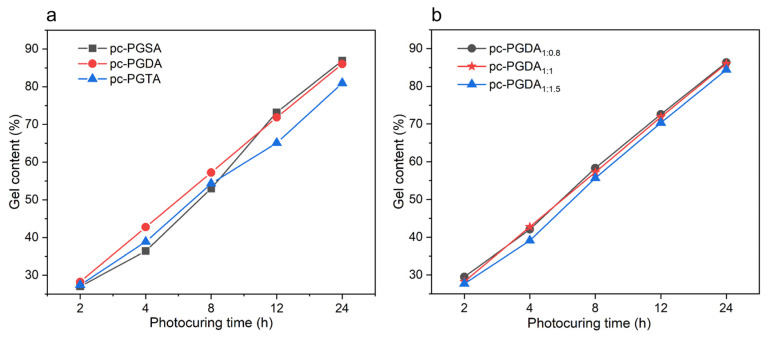
Photocuring curves of the acrylated polyesters. (**a**) Effect of the alkyl chain length of the ADCAs and (**b**) the effect of the Gly: ADCA molar ratios on the photopolymerization times of the acrylated polyesters.

**Figure 5 polymers-16-01278-f005:**
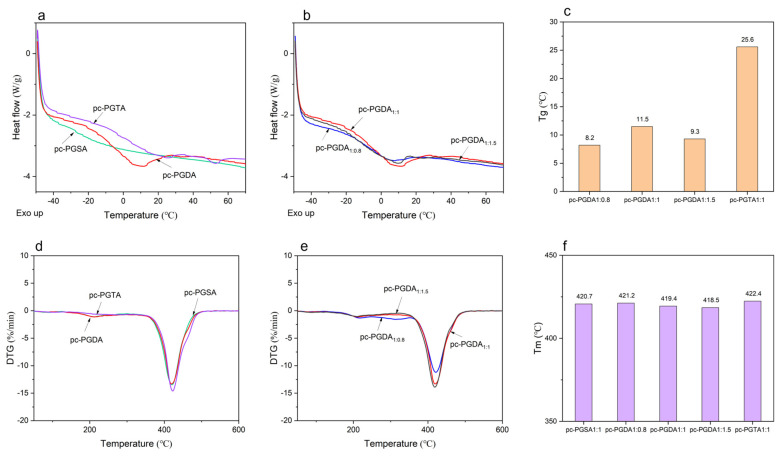
(**a**,**b**) The representative differential scanning calorimetry (DSC) curves of the photocured elastomers; (**c**) the *T*_g_ values of the elastomers; (**d**,**e**) the representative derivative TG (DTG) curves of the photocured elastomers; and (**f**) the *T*_m_ values of the elastomers.

**Figure 6 polymers-16-01278-f006:**
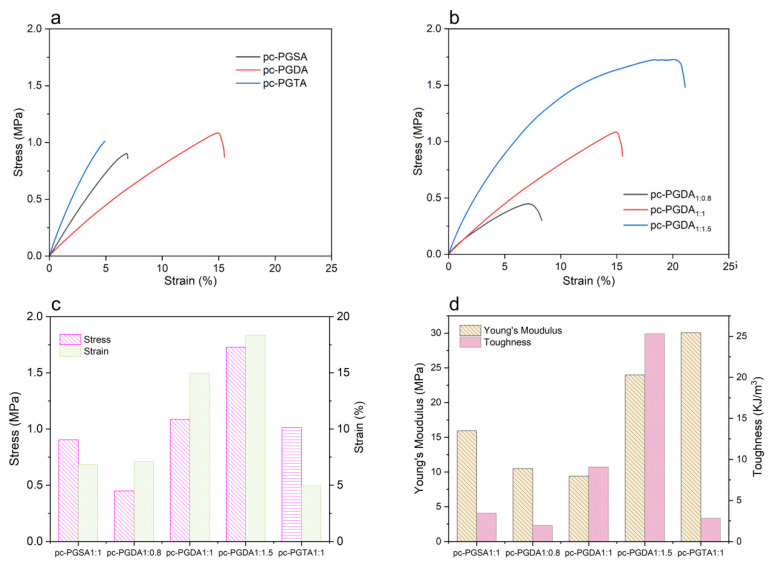
The mechanical properties of the photocured elastomers. (**a**,**b**) The representative stress–strain curves of the photocured elastomers; (**c**) the stress and strain at break of the elastomers; and (**d**) the Young’s modulus and toughness of the elastomers.

**Figure 7 polymers-16-01278-f007:**
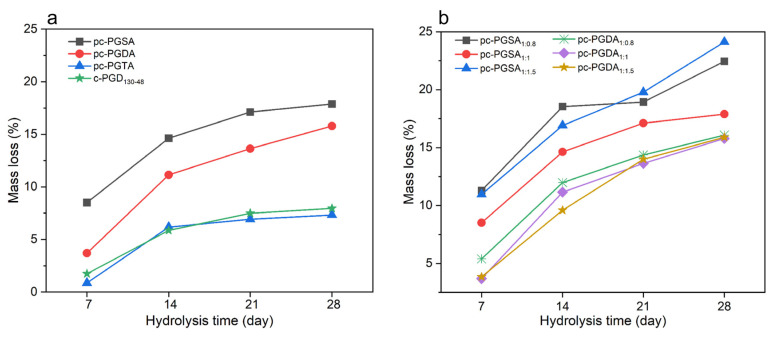
The mass loss profile via hydrolysis of the photocured elastomers. (**a**) The effect of the alkyl chain length of the ADCAs and (**b**) the effect of the Gly: ADCA molar ratios on the weight loss (%) of the elastomers immersed in PBS for 7, 14, 21, or 28 days at 37 °C.

**Table 1 polymers-16-01278-t001:** The molecular weight and *p*-OH/*s*-OH of the prepolymers prepared under the reaction conditions of 120 °C at an N_2_ atmosphere.

Prepolymers	PGS		PGD						PGT	
ADCA	Sebacic acid		Dodecanedioic acid						Tetradecanedioic acid	
Gly:ADCA	1:1		1:0.8		1:1		1:1.5		1:1	
Reaction time (h)	24	48	24	48	24	48	24	48	24	48
*M_n_* (g/mol)	325	410	291	394	346	475	328	505	407	599
*M_w_*(g/mol)	1707	2356	1782	1813	1865	2712	1937	2954	2188	5695
*PDI*	5.25	5.75	6.12	4.60	5.39	5.71	5.91	5.85	5.38	9.51
*p*-OH/*s*-OH	0.92	0.85	1.05	1.02	0.85	0.77	0.73	0.69	0.86	0.70
ED_1_	0.43	0.45	0.32	0.37	0.39	0.44	0.43	0.47	0.39	0.43
ED_2_	0.23	0.27	0.16	0.19	0.23	0.26	0.28	0.29	0.22	0.26
–C=O ^a^	1.83		1.41		1.80		2.55		1.77	

^a^ The quantification of –C=O (δ 172.0–175.0 ppm) was based on the assumption that the carbon region of Gly (δ 62.5–72.5 ppm) in the ^13^C NMR spectra contains 300 carbon atoms. Results are expressed per Gly.

## Data Availability

Data are contained within the article. The raw data supporting the conclusions of this article will be made available by the authors on request.

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
