# Peer review of "Synthesis and Properties of Photocurable Polymers Derived from the Polyesters of Glycerol and Aliphatic Dicarboxylic Acids"

_polymers, 2024, doi:10.3390/polym16091278_

Round 1

Reviewer 1 Report

Comments and Suggestions for Authors

The work is devoted to the synthesis of photocurable polyesters based on glycerol and aliphatic dicarboxylic acids, as well as the study of their properties. In this work, the resulting polyesters are studied in detail using DSC, IR, NMR and GPC methods. Also, the resulting polymers are examined using physical and mechanical methods of analysis and their degradation is analyzed. The work is well written, the data is representative, the authors explain the results obtained and support their conclusions with literature data. The work is of interest for the development of photocurable polymers and their further application, including for the medical field.

During the review of the work, the following comments were made:

1. Figure 1. The reaction diagram should be made larger. It might be worth splitting Figure 1a and Figures 1b and c into two separate figures and making them larger.

2. Was any catalyst used in the synthesis of polyglycerol esters of dicarboxylic acids?

3. Was additional purification of the obtained esters from unreacted reagents carried out?

4. How was the water released during the reaction removed?

5. What method was used to determine the number of hydroxyl groups in molecules?

6. Figure 2. The spectra should be made larger, the positions of the peaks and chemical formulas are difficult to distinguish.

Reviewer 2 Report

Comments and Suggestions for Authors

Reference paper: polymers-2933487

I have read the article entitled “Synthesis and properties of photocurable polymers derived from the polyesters of glycerol and aliphatic dicarboxylic acids” and my comments are summarized below.

Major comment: The authors describe the use of acryloyl chloride to react with the hydroxyl groups in polyesters that are synthesized via polycondensation of glycerol and aliphatic dicarboxylic acids for introducing photoreactive double bonds into the backbone in order to satisfy the rapid photocuring process. Different characterizations using different tools have been carried out. Nevertheless, further information and modifications are necessary to improve the scientific content of the article. The paper could be published with major revisions.

1– While the English grammar is acceptable, a thorough read-through would improve the paper.

2– The abstract needs to be improved, where some important results should be included, to well summarize the content of the paper.

3– The different spectra should be normalized to well see the difference in Fig. 1.

4– In the DSC analysis of the melting process, Add the sigh Endo/Exo in the figure.

5–I can see different processes before 60 °C, does this mean there are several melting processes or there are other phenomena that occur?

6– To further support what is obtained in Figure 7, I suggest adding some TGA experiments to well see the degradation of the different samples.

7––Conclusions can be improved accordingly.

Comments on the Quality of English Language

 Minor editing of English language required

Round 2

Reviewer 2 Report

Comments and Suggestions for Authors

The authors have revised the manuscript according to the reviewer comments, and the paper can be acceptable for publication in its present form.

Comments on the Quality of English Language

Minor editing of English language required